# Evaluation of the Lebanese Adults’ Knowledge Regarding Autism Spectrum Disorder

**DOI:** 10.3390/ijerph20054622

**Published:** 2023-03-06

**Authors:** Melissa Rouphael, Perla Gerges, Christian Andres, Yonna Sacre, Tania Bitar, Walid Hleihel

**Affiliations:** 1Department of Biology, Faculty of Arts and Sciences, Holy Spirit University of Kaslik, Jounieh P.O. Box 446, Lebanon; 2UMR Inserm 1253 Ibrain, Université de Tours, 37032 Tours, France; 3Department of Nutrition and Food Sciences, Faculty of Arts and Sciences, Holy Spirit University of Kaslik, Jounieh P.O. Box 446, Lebanon

**Keywords:** Autism Spectrum Disorder (ASD), autism knowledge, public awareness, general population

## Abstract

The daily functioning and overall well-being of people with ASD depends largely on understanding how the wider public views ASD. Indeed, an increased level of ASD knowledge in the general population may result in earlier diagnosis, earlier intervention, and better overall outcomes. The present study aimed to examine the current state of ASD knowledge, beliefs, and sources of information in a Lebanese general population sample, to identify the factors that could influence this knowledge. A total of 500 participants were involved in this cross-sectional study, which was conducted in Lebanon between May 2022 and August 2022 using the Autism Spectrum Knowledge scale, General Population version (ASKSG). Overall, the participants’ understanding of autism spectrum disorder was low, with a mean score of 13.8 (6.69) out of 32, or 43.1%. The highest knowledge score was found for items related to knowledge of the symptoms and associated behaviors (52%). However, the level of knowledge regarding the etiology and prevalence, assessment and diagnosis, treatment, outcomes, and prognosis of the disease was low (29%, 39.2%, 46%, and 43.4%, respectively). Moreover, age, gender, place of residence, sources of information, and ASD case were all statistically significant predictors of ASD knowledge (*p* < 0.001, *p* < 0.001, and *p* = 0.012, *p* < 0.001, *p* < 0.001, respectively). The general public in Lebanon perceive a lack of awareness and insufficient knowledge regarding ASD. This results in delayed identification and intervention, leading to unsatisfactory outcomes in patients. Raising awareness about autism among parents, teachers, and healthcare professionals should be a top priority.

## 1. Introduction

Autism Spectrum Disorder (ASD), a lifelong neurodevelopmental disorder, has now become a pre-eminent public health concern. It is characterized by persistent deficits in social communication and social interaction, as well as the presence of restricted, repetitive patterns of behavior and interests [1].

The number of ASD patients has dramatically increased over time, with a higher prevalence in boys than girls [2]. There are several potential explanations for this increased prevalence, including improvement in diagnostic criteria, better identification and screening methods, better access to healthcare, and increased awareness among parents and clinicians [3]. Current estimates indicate that 1 in 100 children is diagnosed with ASD around the world [4]. However, this prevalence is substantially higher in Lebanon, where it is estimated to affect between 49 and 513 children per 10,000 across all Lebanese regions [5]. This rise suggests that many people are likely to interact with autistic individuals. Therefore, people must be well-informed, since ignorance increases the risk of misdiagnosis, which makes a child more challenging and therapy-resistant [6]. Hence, it is crucial that the general public have accurate knowledge regarding ASD, since awareness and understanding could lead to early detection and diagnosis of this disorder [7].

Managing autistic children involves the implementation of individualized programs which include behavioral, educational, and psychological interventions. Early access to these ASD interventions depends on the early identification of this disorder and plays a considerable role in outcomes and the improvement in the behavior in the child [8]. Indeed, several studies have shown that early interventions in ASD are linked to significant improvements in cognition, language, and adaptive behavior [9,10].

Parents and teachers have been the main focus of much recent research concerning knowledge and attitudes towards ASD [6,11]. However, limited research has been conducted on the general population’s knowledge of ASD, especially in Lebanon. Golson et al. (2022) assessed the level of autism knowledge among a general population sample in the United States and found that participants in the study were more knowledgeable about the symptoms and behaviors associated with autism than the etiology, prevalence, and assessment procedures [12]. Another study conducted by Alyami et al. (2022) determined the general awareness regarding ASD in Saudi Arabia and assessed variables associated with an accurate understanding of ASD. Overall, the participants showed a weak level of knowledge about ASD, with a mean score of 5.9. Furthermore, participants were more aware about the symptoms and behavior of ASD children, whereas they had a moderate to poor level of knowledge about ASD treatment and etiology [13]. In sum, these results highlight the urgent need for increased awareness of ASD in the general population since it helps in minimizing possible associated social stigma and it is also needed for planning adequate autism education and awareness campaigns for a better inclusion of the ASD patients in the society [14]. 

Therefore, the present study aimed to examine the current state of ASD knowledge, beliefs, and sources of information in a Lebanese general population sample, to identify the factors that could influence this knowledge, and finally to determine the gap in awareness and the lack of information among the general population.

## 2. Materials and Methods

### 2.1. Study Design and Recruitment

Our study was cross-sectional and conducted in all districts in Lebanon, targeting adults aged above 18 years. The number of participants was calculated based on the formula of Krejcie and Morgan (1970) cited by Goyette (2015), given as follows [15]:(1)$$n=\frac{X^{2}NP\left(1-P\right)}{d^{2}\left(N-1\right)+X^{2}P\left(1-P\right)}$$
where *n* = sample size, *X*^2^ = the table value of the chi-squared statistic for one degree of freedom at the desired 5% level of significance = 3.841, *N* = the population size, *P* = the population proportion (0.5), and *d* = the degree of significance (0.05). 

This gave us a sample of 384. Assuming a 20% nonresponse rate, we increased the sample size to 480. However, 500 participants were included in this study to get a better representation of knowledge among the population. 

The data collection started in May 2022 and ended in August 2022. Participation in the research was anonymous and voluntary, with the possibility of withdrawal at any time. The developed questionnaire was uploaded to Google Forms and distributed using social media platforms (WhatsApp and Facebook). It was received after completion and then analyzed statistically. The study description and aims were explained at the beginning of the questionnaire. 

### 2.2. Data Collection Instrument

The data were collected using a questionnaire developed by our team based on the findings of available studies on ASD knowledge [12,13]. It was initially developed in English and subsequently translated into Arabic to ensure that the Lebanese participants were able to understand it. A pilot study was conducted on 25 Lebanese adults to assess the clarity and comprehension of the questionnaire. After the analysis of the pilot study, our research team revised, updated, and submitted the questionnaire to the ethical committee of USEK. who reviewed and approved it. The questionnaire consisted of three sections: socio-demographic characteristics, ASD knowledge, and general beliefs about ASD. 

#### 2.2.1. Socio-Demographic Characteristics

Participants were asked to report their gender and age, as well as other information such as marital status, education level, and where they live. 

#### 2.2.2. ASD Knowledge

Participants were asked to report their rate of understanding of ASD and list where their knowledge came from. They were also asked whether they knew an autistic patient and whether they were aware of any special centers for children with ASD in Lebanon. Moreover, they were asked to define ASD.

#### 2.2.3. Beliefs about ASD

The participants were also given the Autism Spectrum Knowledge Scale, General Population Version (ASKSG), a validated questionnaire specifically designed for the general population and edited by our team in order to meet the objectives of our study (McClain et al., 2019). It contains five domains and is comprised of 32 items: (1) etiology and prevalence (7 items), (2) symptoms and behaviors (12 items), (3) assessment and diagnosis (5 items), (4) treatment (3 items), and (5) outcomes and prognosis (5 items) [12]. Participants must answer one of the following options: “true”, “false”, or “don’t know”. Each correct answer was given a score of 1, or a score of 0 if the response was not provided correctly. A score of 0 was also given for a response of “don’t know” on the assumption that this response implies a lack of knowledge about the item. The higher the score, the more knowledgeable the participant. Therefore, a maximum and minimum total score of 32 and 0, respectively, is possible when the five domain scores are added together. The items are included in Table 1.

### 2.3. Ethical Considerations

The study complied with the ethical standards and guidelines of the Declaration of Helsinki of 1964 and its later amendments. The Holy Spirit University of Kaslik Ethical Committee reviewed and approved the study protocol (EC 90010141). This study did not result in any physical or psychological harm to the participants. Moreover, the credentials of the participants were kept confidential, and the results of the study were anonymous and used only for academic purposes.

### 2.4. Data Analysis

The data for this study were analyzed using SPSS version 20.0 (IBM SPSS Statistics 20). Descriptive statistics were used to describe the demographic characteristics and the previous knowledge of ASD of the study population. Individual correct responses for each domain were counted and displayed using percentages. The percentage of the knowledge score in each domain and the total score were calculated and categorized using a modified Bloom’s cut-off, as good if the score was between 75 and 100%, moderate if the score was between 50 and 75%, and low if the score was less than 50% [1]. The knowledge scores of different participants according to demographic characteristics were compared using the chi-square test. Multivariable linear regression analysis using all the demographic variables as independent variables and knowledge score as the outcome variable was conducted to identify predictors associated with this knowledge. A *p*-value < 0.05 was considered statistically significant.

## 3. Results

### 3.1. Participants’ Demographic Characteristics

A total of 500 participants, with 68.8% female, were involved in the study. The sample had a variable age distribution, with 53.2% being aged above 35 years. Around half of the participants were single (47.6%) and half were married (47.4%). Most participants were educated (77.8% at university level) and lived in Mount Lebanon (68.2%). The demographic characteristics of the study population are summarized in Table 2.

### 3.2. Perceived Knowledge among Participants and Their Source of Information

When asked if they had heard about ASD, 88.8% responded positively. However, when asked to rate their degree of knowledge, only 8% felt that they were familiar with autism. The main source of their information was social media (54.34%) (Figure 1).

Moreover, 45.8% were aware of specialized centers for children with ASD in Lebanon. A total of 65.2% stated that they did not know someone who had ASD, while 34.8% knew someone who had ASD. Among these, 17.14% had a family member with autism, 37.14% had a friend with autism, and 45.7% had a colleague with autism. Furthermore, 44.8% defined autism as a behavioral and social disorder, 33.8% as a neurodevelopmental disorder, 15.2% as a psychiatric disorder, and 6.2% as an emotional disorder (Table 3).

### 3.3. Actual Knowledge about ASD

#### 3.3.1. Current State of ASD Knowledge and Beliefs

Participant knowledge was evaluated in terms of its content area (Table 4). Regarding knowledge of ASD etiology and prevalence, 47.6% endorsed the myth that vaccines cause ASD. Only 35% agreed that ASD is more common among boys than girls, and 53.2 percent identified that the emotional deprivation of the mother was not a cause of ASD. Few participants mentioned that advanced paternal or maternal age is a risk factor for ASD (21% and 11.4%, respectively). Moreover, few participants (17.4%) knew that there are racial/ethnic/regional disparities in ASD identification, and only 17.4% agreed that children with siblings who have ASD are at a higher risk of developing the disorder.

Most of the participants (68.8%) disagreed that all individuals with ASD have a low intellectual quotient (IQ) and 71% mentioned that children with ASD may not play with things the way they are intended. Most of the participants demonstrated a strong understanding of the symptoms of ASD: the presence of strict routines or rituals (85.4%), difficulties interacting socially with others (63.8%), difficulties expressing themselves (72.6%), difficulties in coordination (78%), and the presence of feeding problems (57.4%). Few of the participants disagreed that all individuals with ASD display aggressive behaviors and knew that symptoms of ASD appear before the age of 2 years (16.60% and 20.4%, respectively). Moreover, a few of the participants indicated that children with ASD experience sleep disturbances, gastrointestinal problems, and epilepsy (14%, 30%, and 27.4%, respectively).

Of the participants, 65.8% agreed that diagnosis of ASD is primary based on behavioral observations and parents’ interviews, while 42.8% considered that ASD can only be diagnosed after the age of 4 years. Few participants disagreed that ASD can be diagnosed using only brain imaging (37.6%) and it can develop in adulthood (10.6%). Only 39.8% knew that symptoms may occur from early childhood.

Moreover, 41.8% stated that there are beneficial interventions for individuals with ASD. In fact, most of the participants (79.2%) considered social skills training to be one of the most effective interventions for these individuals. However, only 16.8% knew that restricting certain foods and giving supplements was an effective intervention for ASD.

Regarding knowledge of autism outcomes and prognosis, most of the participants (73.8%) agreed that individuals with ASD can learn to speak and exhibit difficulties living and working independently in adulthood (51.8%). Half of the participants (56.8%) knew that individuals with ASD also have additional mental health issues such as anxiety, depression, obsessive-compulsive disorder (OCD), and attention deficit hyperactivity disorder (ADHD). Most of the participants were unsure and responded “ I don’t know” when asked if symptoms of ASD, such as delayed language and movement skills, and metabolic syndromes such as mitochondrial disorder and creatine deficiency, remain stable throughout an individual’s life (25% and 11.4%, respectively).

#### 3.3.2. Participants’ Knowledge Score and Their Predictors

##### Total Scores and Subscales

The mean score on the questionnaire across all participants was 13.8 (SD = 6.69), indicating overall low-level knowledge about autism spectrum disorder. The highest knowledge score was found for items related to knowledge of the symptoms and associated behaviors (52%). However, the level of knowledge regarding the etiology and prevalence, assessment and diagnosis, treatment, and outcomes and prognosis of the disease was low (29%, 39.2%, 46%, and 43.4%, respectively), (Table 5) and (Figure 2).

##### Factors Associated with ASD Knowledge

To determine the factors that influence ASD knowledge, the autism knowledge score was stratified by demographic characteristics (age, gender, marital status, region of residence, and highest education level) as shown in Table 6. Knowledge scores differed significantly by gender (*p* < 0.001), age group (*p* = 0.012), and region of residence (*p* = 0.026). In fact, more participants in the 25–34 years age group showed a good understanding of ASD compared to the other age groups; females showed a good understanding of ASD compared to males; and participants living in Beirut had significantly more knowledge of ASD compared to those of other regions. Moreover, our study found that there was a significant difference in knowledge between people who had a colleague with autism and those without (*p* < 0.001), and a significant difference was found between ASD knowledge and social media (*p* < 0.001), Table 6.

Five socio-demographic variables (gender, age, region of residence, source of information, and known ASD case) showed a significant association with the respondents’ knowledge using univariate logistic regression. These five variables were analyzed using a multivariate logistic regression to determine their independent influence on ASD knowledge; Table 7.

Age, gender, region of residence, source of information, and ASD case were all statistically significant predictors of ASD knowledge (*p* < 0.001, *p* < 0.001, and *p* = 0.012, *p* < 0.001, *p* < 0.001, respectively).

## 4. Discussion

To the best of our knowledge, this study is the first to examine the current state of ASD knowledge and beliefs in the Lebanese general population. The daily functioning and overall well-being of people with ASD depends largely on understanding how the wider public views ASD. Indeed, an increased level of ASD knowledge in the general population may result in earlier diagnosis, earlier intervention, and better overall outcomes.

In our study, the vast majority of the study participants reported that they had prior knowledge about autism spectrum disorder; however, wide gaps were identified in their perceived knowledge. This confirmed the findings of a previous Australian study, which found that despite having heard about ASD, participants had a poor understanding of the disorder and had many misconceptions about it [2]. Moreover, participants showed a low level of knowledge about ASD. The highest knowledge score was found for items related to symptoms and associated behaviors. However, the level of knowledge about the cause, prevalence, diagnosis, outcomes, and treatment of this disorder was low. These findings contrast with a previous study conducted in the United States, which found that most participants demonstrated the most knowledge regarding the symptoms and behaviors associated with autism [3]. Participants answered an average proportion of 64.3% of items correctly on the ASKSG, but in this study the proportion was lower (43.1%) [12]. In a previous study conducted in Saudi Arabia, the participants had a poor understanding of the disorder and many misconceptions about it. There was a knowledge gap in the community regarding ASD, particularly among males, which is similar to the results of this study [13]. Knowing the risk factors for ASD is important to reduce the chances of having a child with this disorder. In fact, it is proposed that several environmental factors could contribute to the development of this disorder. Among these have been suggested advanced maternal age (35 years), maternal chronic hypertension, pre-eclampsia, gestational hypertension, and being overweight before or during pregnancy [16]. A case-control study, conducted between 2015 and 2020 in the Lebanese population, shed light on risk and protective factors associated with ASD. In this study, consanguinity, a familial history of ASD and attention deficit hyperactivity disorder (ADHD), and maternal stress during pregnancy were identified as risk factors associated with ASD [17]. Moreover, early detection of the disorder can significantly improve the quality of life of individuals with ASD as well as that of their caregivers and families [9].

Social media was the most popular source of knowledge about ASD among our participants, followed by personal experience, books, doctors, and other sources. Social media is becoming a valuable platform for facilitating knowledge sharing and communication [18]. For example, films are being used for educational and entertaining purposes. One study conducted by Conn et al. (2012) assessed the impact of 23 Hollywood films that portray autism spectrum disorders on public understanding as well as on medical students and psychiatric trainees. They found that these films provide the best opportunity for medical education and improvement in public awareness [19]. At the same time, there are rising concerns that these social media platforms are spreading inaccurate or misleading health information. Indeed, misinformation about health is prevalent on social media, and is frequently defined as information that disputes the findings of experts in the field [20]. As a result, information seekers must search for medical information from reliable resources such as the WHO or the Centers for Disease Control (CDC) [21].

Furthermore, we found that females had the highest percentages of accurate responses. In fact, studies that evaluated knowledge by gender indicated that females are more knowledgeable about ASD than males. One possible explanation for this is that females are more interested in studying diseases than males [14,22]. A previous study that examined the knowledge about ASD in China reported that gender and socioeconomic status were important variables that had an influence on ASD knowledge [23]. Moreover, our results showed that there was a significant difference in knowledge between people who had a colleague with autism and those without. This finding is consistent with earlier research. According to earlier studies, participants who directly interacted with people with ASD knew more since they had more experience [24].

According to the current study, ASD deserves more attention. In fact, this study highlights the need to educate the Lebanese population about this disorder. First, a large number of participants agreed with myths regarding the causes of ASD (such as vaccine etiology), its diagnosis (such as the age of diagnosis), and treatment (such as restrictive diets). Continued belief in the vaccine myth is alarming as it lowers rates of vaccination and endangers both individual and public health [25]. Misconceptions regarding ASD services are also concerning. Families may waste money on ineffective interventions rather than effective services. Lack of awareness about ASD diagnosis and identification may contribute to delays in age of diagnosis [26]. Increased ASD detection, parent psychoeducation, and communication between healthcare and parents are essential to addressing this knowledge gap.

Second, while most people are aware of the symptoms and behaviors associated with ASD, less is known about the assessment and diagnosis. Increased public knowledge regarding this domain could ensure early detection. A variety of strategies could be implemented to raise awareness, assist in early diagnosis, and improve patient outcomes. Social media campaigns may be a useful strategy to increase community understanding of ASD. Public organizations such as schools, community centers, and clinics should prioritize public awareness campaigns such as fliers, informational sessions, and advertisements on social media, or in newspapers, radio, and television. Therefore, we will launch an awareness-raising campaign about ASD in schools, universities, and community centers to help in early recognition and enhance public awareness.

All of these mentioned approaches aim to increase the quality of life of individuals with ASD and move from awareness to acceptance and inclusion. In fact, children with ASD have the right to be in inclusive classrooms, and participate and be accepted in society. The concept of inclusion means that students with autism should be educated in the same environment as typically developing students with appropriate support services [27]. Harrower et al. (2001) documented that students with disabilities, including students with autism, who are fully included in a general classroom environment displayed higher levels of engagement and social interaction. Moreover, they had larger friendship networks and received higher levels of social support [28].

This study has limitations that must be pointed out. First, the sample was not fully representative as most participants lived in Mount Lebanon. Thus, results cannot be generalized to the whole country. Second, the instrument used was a self-administered questionnaire based on participant responses. This may raise questions of quality since participants may answer quickly, dishonestly, or inattentively. Third, the sample size for the general public was small. A bigger sample is needed for future study.

## 5. Conclusions

In conclusion, the general public lack awareness and have insufficient knowledge regarding ASD. This results in delayed identification and intervention, leading to unsatisfactory outcomes in patients. Raising awareness about autism among parents, teachers, and healthcare professionals should be a top priority. Some participants’ characteristics, including gender, age, and education level, were found to be associated with better ASD knowledge. To fill this knowledge gap and ensure early detection, appropriate diagnosis, and evidence-based interventions, specific and focused strategies must be implemented. This includes awareness campaigns, fliers, informational sessions, and advertisements on social media, or in newspapers, radio, and television. Further studies targeting early childhood educators, medical students, pediatricians, parents who have ASD children, teachers, and nurses should be conducted to raise awareness and knowledge.

## Figures and Tables

**Figure 1 ijerph-20-04622-f001:**
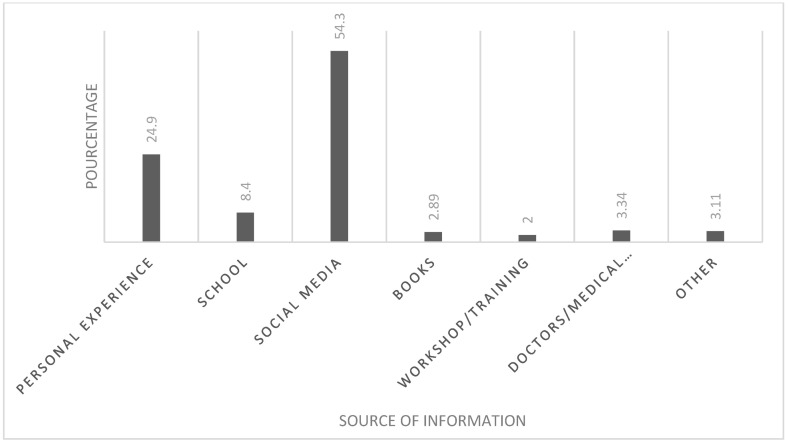
Participants’ sources of information about ASD.

**Figure 2 ijerph-20-04622-f002:**
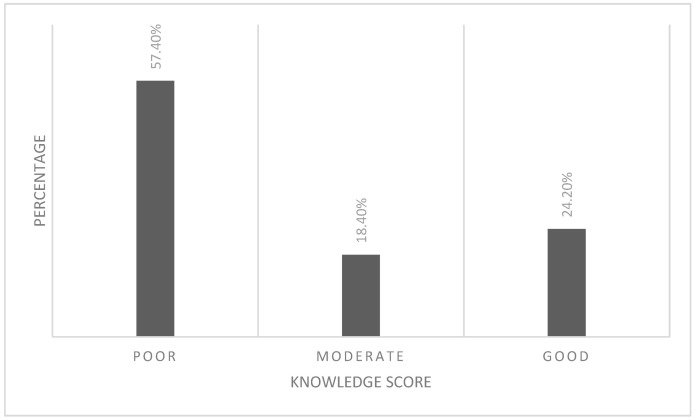
Participant’s knowledge score. Knowledge score < 50% = poor knowledge, 50–75% score = moderate knowledge, >75% score = good knowledge.

**Table 1 ijerph-20-04622-t001:** Items and their responses.

Item		Response
1	Vaccines can cause ASD	False
2	ASD is more common among boys than girls	True
3	ASD is caused by emotional deprivation from the mother	False
4	Children with siblings who have ASD are at a higher risk of developing the disorder	True
5	Advanced maternal age is a risk factor of ASD	True
6	Advanced paternal age is a risk factor of ASD	True
7	There are no differences in the identification rate of ASD across regional, racial and ethnic groups	False
8	All individuals with ASD have low intellectual quotient (IQ)	False
9	Children with ASD may not play with things the way they are intended	True
10	Children with ASD may have strict routines or rituals	True
11	Individuals with ASD have difficulties interacting socially with others	True
12	Individuals with ASD may be uncoordinated or clumsy	True
13	Individuals with ASD have difficulties expressing themselves	True
14	Symptoms of ASD do not appear before the age of 2 years	False
15	Diagnosis of ASD is primarily based on behavioral observations and parents’ interviews	True
16	ASD can only be diagnosed after the age of 4 years	False
17	ASD can be diagnosed using brain imaging only	False
18	For a diagnosis of ASD, symptoms must be present from early childhood	True
19	It is possible for ASD to develop in adulthood	False
20	There are no beneficial interventions for individuals with ASD	False
21	Restricting certain food (e.g., gluten) and giving supplements (vitamin D, magnesium…) are an effective intervention for ASD	True
22	Social skills (visual support, role-play, video-modelling…) training is an effective intervention for individuals with ASD	True
23	Individuals with ASD can learn to speak	True
24	Individuals with ASD have difficulties living and working independently in adulthood	True
25	Up to 70% of individuals with ASD also have an additional mental health such as anxiety, depression, obsessive-compulsive disorder OCD, attention Deficit Hyperactivity Disorder ADHD	True
26	All individuals with ASD display aggressive behavior	False
27	Individuals with ASD experience sleep disturbances	True
28	Individuals with ASD experience feeding problems	True
29	Individuals with ASD experience gastrointestinal problems	True
30	Individuals with ASD experience epilepsy	True
31	After being diagnosed and treated, symptoms of ASD such as delayed language and movement skills, learning skills remain stable throughout an individual’s life	False
32	After being diagnosed and treated, metabolic syndromes such as mitochondrial disorder, creatine deficiency, purine metabolism disorder remain stable throughout an individual’s life	False

**Table 2 ijerph-20-04622-t002:** Participants’ demographic characteristics.

Variable	N	Percentage (%)
Gender		
Female	343	68.6
Male	148	29.6
Prefer not to say	9	1.8
Age		
18–24	114	22.8
25–34	120	24
35–49	169	33.8
50 and above	97	19.4
Marital status		
Single	238	47.6
Married	237	47.4
Divorced	7	1.4
Widowed	12	2.4
Prefer not to say	6	1.2
Highest education level		
Primary education	3	0.6
Intermediate education	20	4
Secondary education	84	16.8
Bachelor degree	154	30.8
Master degree	202	40.4
Doctorate	33	6.6
No degree	2	0.4
Prefer not to say	2	0.4
Region of residence		
Akkar	4	0.8
North	64	12.8
Baalback	7	1.4
Bekaa	15	3
Mount Lebanon	341	68.2
Beirut	51	10.2
Nabatieh	1	0.2
South	14	2.8
Prefer not to say	3	0.6

**Table 3 ijerph-20-04622-t003:** Perceived knowledge about ASD.

Variable	N	Percentage (%)
Have you heard about Autism Spectrum Disorder?		
Yes	444	88.8
No	56	11.2
If yes, how would you rate your understanding of ASD?		
Very little understanding	211	42.2
Some understanding	199	39.8
Very familiar with autism	40	8
Are you aware about any special centers for children with ASD in Lebanon?		
Yes	229	45.8
No	271	54.2
Do you know someone who has ASD?		
Yes	174	34.8
No	326	65.2
If yes, please indicate the relationship		
Family member	30	17.14
Friends’ family	65	37.14
Colleagues’ family	80	45.71
How would you define ASD?		
Psychiatric disorder	76	15.2
Emotional disorder	31	6.2
Behavioral/social disorder	224	44.8
Neurodevelopmental disorder	169	33.8

**Table 4 ijerph-20-04622-t004:** Participants’ responses for knowledge items.

Item	True Response N (%)	False Response N (%)
Domain 1: Etiology and prevalence		
Vaccines can cause ASD	238 (47.6%)	262 (52.4%)
ASD is more common among boys than girls	175 (35.0%)	325 (65.0%)
ASD is caused by emotional deprivation from the mother	266 (53.2%)	234 (46.8%)
Children with siblings who have ASD are at a higher risk of developing the disorder	87 (17.4%)	413 (82.6%)
Advanced maternal age is a risk factor of ASD	105 (21.0%)	395 (79.0%)
Advanced paternal age is a risk factor of ASD	57 (11.4%)	443 (88.6%)
There are no differences in the identification rate of ASD across regional, racial and ethnic groups	87 (17.4%)	413 (82.6%)
Domain 2: Symptoms and associated behaviors		
All individuals with ASD have low intellectual quotient (IQ)	344 (68.8%)	156 (31.2%)
Children with ASD may not play with things the way they are intended	356 (71.2%)	144 (28.8%)
Children with ASD may have strict routines or rituals	427 (85.4%)	73 (14.6%)
Individuals with ASD have difficulties interacting socially with others	319 (63.8%)	181 (36.2%)
Individuals with ASD may be uncoordinated or clumsy	390 (78.0%)	110 (22.0%)
Individuals with ASD have difficulties expressing themselves	363 (72.6%)	137 (27.4%)
Symptoms of ASD do not appear before the age of 2 years	102 (20.4%)	398 (79.6%)
All individuals with ASD display aggressive behavior	83 (16.6%)	417 (83.4%)
Individuals with ASD experience sleep disturbances	70 (14.0%)	430 (86%)
Individuals with ASD experience feeding problems	287 (57.4%)	213 (42.6%)
Individuals with ASD experience gastrointestinal problems	150 (30%)	350 (70%)
Individuals with ASD experience epilepsy	137 (27.4%)	363 (72.6%)
Domain 3: Assessment and diagnosis		
Diagnosis of ASD is primarily based on behavioral observations and parents’ interviews	329 (65.8%)	171 (34.2%)
ASD can only be diagnosed after the age of 4 years	214 (42.8%)	286 (57.2%)
ASD can be diagnosed using brain imaging only	188 (37.6%)	312 (62.4%)
For a diagnosis of ASD, symptoms must be present from early childhood	199 (39.8%)	301 (60.2%)
It is possible for ASD to develop in adulthood	53 10.6%)	447 (89.4%)
Domain 4: Treatment		
There are no beneficial interventions for individuals with ASD	209 (41.8%)	291 (58.2%)
Restricting certain food (e.g., gluten) and giving supplements (vitamin D, magnesium…) are an effective intervention for ASD	84 (16.8%)	416 (83.2%)
Social skills (visual support, role-play, video-modelling…) training is an effective intervention for individuals with ASD	396 (79.2%)	104 (20.8%)
Domain 5: Outcomes and prognosis		
Individuals with ASD can learn to speak	369 (73.8%)	131 (26.2%)
Individuals with ASD have difficulties living and working independently in adulthood	259 (51.8%)	241 (48.2%)
Up to 70% of individuals with ASD also have an additional mental health such as anxiety, depression, obsessive-compulsive disorder OCD, attention Deficit Hyperactivity Disorder ADHD	284 (56.8%)	216 (43.2%)
After being diagnosed and treated, symptoms of ASD such as delayed language and movement skills, learning skills remain stable throughout an individual’s life	125 (25%)	375 (75%)
After being diagnosed and treated, metabolic syndromes such as mitochondrial disorder, creatine deficiency, purine metabolism disorder remain stable throughout an individual’s life	57 (11.4%)	443 (88.6%)

**Table 5 ijerph-20-04622-t005:** Scores overall and subscales.

Scale	Number of Items	Mean (SD)	Percentage (%)	Level of Knowledge
Knowledge about etiology and prevalence	7	2.03 (1.63)	29	Poor
Knowledge about symptoms and associated behaviors	12	6.24 (2.91)	52	Moderate
Knowledge about assessment and diagnosis	5	1.96 (1.42)	39.2	Poor
Knowledge about treatment	3	1.38 (0.858)	26	Poor
Knowledge about outcomes and prognosis	5	2.17 (1.35)	43.4	Poor
Total score for all items	32	13.8 (6.69)	43.1	Poor

**Table 6 ijerph-20-04622-t006:** Comparison of knowledge scores according to socio-demographic variables.

Variables		Knowledge		
		Poor Knowledge	Moderate Knowledge	Good Knowledge
		N (%)	N (%)	N (%)
Gender	Female	170 (34%)	75 (15%)	98 (19.6%)
	Male	111 (22.2%)	15 (3%)	22 (4.4%)
	Prefer not to say	6 (1.2%)	2 (4.6%)	1 (0.2%)
*p*-value	<0.001 *			
Age	18–24	64 (12.8%)	19 (3.65%)	31 (6.2%)
	25–34	55 (11%)	23 (4.6%)	42 (8.4%)
	35–49	103 (20.6%)	35 (7%)	31 (6.2%)
	50 and above	65 (13%)	15 (3%)	17 (3.4%)
*p*-value	0.012 *			
Marital status	Single	138 (27%)	41 (8.2%)	59 (11.8%)
	Married	133 (26.6%)	45 (9%)	59 (11.8%)
	Divorced	5 (1%)	2 (0.4%)	0 (0%)
	Widowed	9 (1.8%)	1 (0.2%)	2 (0.4%)
	Prefer not to say	2 (0.4%)	3 (0.6%)	1 (0.2%)
*p*-value	0.404			
Region of residence	North	1 (0.2%)	1 (0.2%)	2 (0.4%)
	Baalback	27 (5.4%)	10 (2%)	27 (5.4%)
	Bekaa	5 (1%)	1 (0.2%)	1 (0.2%)
	Mount Lebanon	12 (2.4%)	1 (0.2%)	2 (0.4%)
	Beirut	205 (41%)	66 (13.2%)	73 (14.6%)
	Nabatieh	32 (6.4%)	8 (1.6%)	11 (2.2%)
	South	0 (0%)	0 (0%)	1 (0.2%)
	Prefer not to say	5 (1%)	5 (1%)	4 (0.8%)
*p*-value	0.026 *			
Highest education level	Primary education	2 (0.4%)	1 (0.2%)	0 (0%)
	Intermediate education	14 (2.8%)	4 (0.8%)	2 (0.4%)
	Secondary education	58 (11.6%)	13 (2.6%)	13 (2.6%)
	Bachelor degree	90 (18%)	23 (4.6%)	41 (8.2%)
	Master degree	110 (22%)	44 (8.8%)	48 (9.6%)
	Doctorate	11 (2.2%)	7 (1.4%)	15 (3%)
	No degree	1 (0.2%)	0 (0%)	1 (0.2%)
	Prefer not to say	1 (0.2%)	0 (0%)	1 (0.2%)
*p*-value	0.062			
ASD case	Family’s member	8 (1.6%)	7 (1.4%)	15 (3%)
	Friends’ member	27 (5.4%)	15 (3%)	23 (4.6%)
	Colleagues’ member	31 (6.2%)	17 (3.4%)	32 (6.4%)
*p*-value	<0.001 *			
Source of information	Personal experience	51 (10.2%)	20 (4%)	41 (8.2%)
	School	22 (4.4%)	9 (1.8%)	11 (2.2%)
	Media	153 (30.6%)	47 (9.4%)	44 (8.8%)
	Books	5 (1%)	5 (1%)	3 (0.6%)
	Workshops	0 (0%)	2 (0.4%)	7 (1.4%)
	Doctor	2 (0.4%)	3 (0.6%)	10 (2%)
	other	5 (1%)	4 (0.8%)	5 (1%)
*p*-value	<0.001 *			

* *p*-value < 0.05.

**Table 7 ijerph-20-04622-t007:** Multiple linear regression with ASD knowledge as the dependent variable.

Independent Variables	*p*-Value	95% Confidence Intervals	
		Lower Limit	Upper Limit
Age	0.000	−3.973	−1.753
Gender	0.012	−1.236	−0.156
Region of residence	0.026	−0.942	−0.062
ASD case	0.000	−2.450	−1.202
Source of information	0.002	−0.712	−0.153

## Data Availability

Not applicable.

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
