# Peer review of "Evaluation of the Lebanese Adults’ Knowledge Regarding Autism Spectrum Disorder"

_ijerph, 2023, doi:10.3390/ijerph20054622_

Round 1
Reviewer 1 Report
The manuscript entitled ‘Evaluation of the Lebanese Adults’ Knowledge Regarding Autism Spectrum Disorder’ by Melissa Roufael et al. represents an interesting manuscript where the authors want to evaluate the state of ASD knowledge, believes and source of information. Additionally, the authors want to identify factors that could influence this knowledge. In overall, the main purpose of the present study is to find the gaps in awareness among the general Lebanese population.
However, the present manuscript form is very preliminary, and more experiments are needed for clearly characterize the experimental groups. Therefore, the present version of the manuscript should not be accepted, and significant changes would be necessary.
In overall, I consider that the premise of this study is interesting and important for the field, and I will perform some comments and suggestions
Major and Minor concerns:
1. The authors referred that study protocol was reviewed and approved by the Holy Spirit University Ethical Committee. The approval number should also be indicated.
2. Who were the persons that made the revision/validation of questionnaire applied to the study participants? This information should be included in methods section.
3. The study design has some flaws, the number of study participants (males and females) these should be proportional. Most of study participants are from Mount Lebanon, why? A higher diversity of locations should be included.
4. The table is very difficult to understand and should be simplified. For example, about ASD how was evaluated the following parameters: very little understanding, some understanding and very familiar with autism??
5. Table 3, 229 persons indicated that were aware about any special centers for children with ASD in Lebanon. But only 225 were listed below? Why?
6. Figure2, 3, 4, 5 and 6 have graph title that are the same of legend. I believe these could be simplified. For instance, Figure 2 title: Etilogy and Prevalence instead of Participants’ responses to knowledge about autism etiology and prevalence.
7. Figure 3 X axis should be improved.
8. How was calculated the Participants knowledge score? ( Knowledge score < 50% = poor knowledge, 50 – 75% score 232 = moderate knowledge,. > 75% score = good knowledge)
9. The Knowledge scores differed significantly 237 by gender (p<0.001). How the authors justified that 343 females and only 149 males were included in the present study? The same happens with the place of living?
10. The manuscript conclusions are too speculative and are not based on results presented so far.
Author Response
We appreciate the time and effort the reviewer has dedicated to providing his valuable feedback on our manuscript.
We are grateful for his insightful comments on our submitted article.
We have been able to incorporate changes to reflect the suggestions provided by the reviewer.
We have highlighted the changes within the manuscript.

Reviewer 2 Report
This is an important and interesting study. I do, however, have a few recommendations which are included in the attached document.

Author Response

(The authors gave the same response as above.)

Round 2
Reviewer 1 Report
Please find my comments about the revision of the manuscript entitled ‘The manuscript entitled ‘Evaluation of the Lebanese Adults’ Knowledge Regarding Autism Spectrum Disorder’ by Melissa Roufael et al. The authors respond to all issues raised in my revision and performed the adequate alterations of the manuscript and the latter was significantly improved.
Overall, I believe that the manuscript is ready for publication.